# Genetic predictors of participation in optional components of UK Biobank

Jessica Tyrrell [1✉], Jie Zheng[2,3], Robin Beaumont[1], Kathryn Hinton[1], Tom G. Richardson [2,3], Andrew R. Wood[1], George Davey Smith [2,3], Timothy M. Frayling [1] & Kate Tilling [2,3✉]

Large studies such as UK Biobank are increasingly used for GWAS and Mendelian randomization (MR) studies. However, selection into and dropout from studies may bias genetic and phenotypic associations. We examine genetic factors affecting participation in four optional components in up to 451,306 UK Biobank participants. We used GWAS to identify genetic variants associated with participation, MR to estimate effects of phenotypes on participation, and genetic correlations to compare participation bias across different studies. 32 variants were associated with participation in one of the optional components ($P < 6 \times 10^{-9}$), including loci with links to intelligence and Alzheimer's disease. Genetic correlations demonstrated that participation bias was common across studies. MR showed that longer educational duration, older menarche and taller stature increased participation, whilst higher levels of adiposity, dyslipidaemia, neuroticism, Alzheimer's and schizophrenia reduced participation. Our effect estimates can be used for sensitivity analysis to account for selective participation biases in genetic or non-genetic analyses.

[1] Genetics of Complex Traits, College of Medicine and Health, University of Exeter, Exeter, UK. [2] MRC Integrative Epidemiology Unit at the University of Bristol, University of Bristol, Bristol, UK. [3] Population Health Sciences, Bristol Medical School, University of Bristol, Bristol, UK. ✉email: j.tyrrell@exeter.ac.uk; Kate.Tilling@bristol.ac.uk

nitial participation and continued engagement in cohort studies may be influenced by an individual's social and lifestyle characteristics[1]. This selection has the potential to result in bias in estimating phenotypic and genotypic associations[2]. It is well established that large cohort studies tend to have a healthy volunteer bias in initial participation[3]. There is also a growing body of evidence suggesting that continued engagement in cohort studies may be influenced by a range of factors. Studies have demonstrated age, education, ancestry, geographic location and health status are associated with loss-to-follow-up[4]. However, as with all observational studies, these associations may be confounded and therefore not causal in nature.

In order to assess the causes of non-participation, genetic data on a more complete sample can be leveraged. Analysis of genetic data in the Avon Longitudinal Study of Parents and Children (ALSPAC) demonstrated a number of factors were causally related to participation. Education, agreeableness and openness caused higher participation, whilst higher BMI, schizophrenia, neuroticism and depression caused lower participation[1,5]. A study in the UK Biobank[6] performed a genome-wide association study of completing the mental health questionnaire, identifying 25 loci associated with survey completion, and strong positive genetic correlations with educational attainment and better health and negative genetic correlations with psychological distress and schizophrenia.

In general, an analysis will give biased estimates if the exposure and the outcome variable (or causes of them) are associated with participation (conditional on the other variables in the analysis model[7]). Selection bias can also occur under other circumstances when only the outcome is related to selection, for example, if exposure does cause the outcome, and the outcome causes selection[7]. As another example, selection bias can occur if a modifier of the effect of exposure on outcome causes selection[8]. A comparison of associations between risk factors and overall and cause-specific mortality in UK Biobank and the less-selected Health Survey for England and Scottish Health Surveys showed wide variation in these associations[9], with some over-estimated in UK Biobank and some under-estimated. Thus, to understand the impact of selective participation for a particular analysis, we need to identify factors that influence participation.

The UK Biobank has several measures of participation. Here, we utilise up to 451,036 individuals of European ancestry in the UK Biobank to identify factors that cause participation in the four available optional components of the baseline study in order to improve our understanding of the biases that may affect these associations and lead to false inferences. The four optional components tested were (a) the percentage of food frequency questionnaires (FFQ) completed, (b) acceptance of the invite to wear a physical activity monitor, (c) acceptance of an invitation to participate in the mental health questionnaire (MHQ) and (d) the completion of the aide-memoire. We used two-sample Mendelian randomisation (MR) approaches to explore the role of over 80 predictors on participation in the UK Biobank. Finally, we also explored genetic correlations between participation in the UK Biobank and the ALSPAC study to test between study consistency. If the same factors affect participation in studies that vary by geography, time period and design, then those studies will suffer the same bias, and thus replication of results across studies becomes meaningless[10].

This study identified 32 variants associated with participation in at least one of the four optional components ($P < 6 \times 10^{-9}$), including loci with known links to intelligence and Alzheimer's disease. Genetic correlations demonstrated that participation bias was common across studies, whilst MR provided evidence that longer educational duration, older menarche and taller stature increased participation, whilst higher levels of adiposity,

dyslipidaemia, neuroticism, Alzheimer's and schizophrenia reduced participation. Our effect estimates can be used for sensitivity analysis to account for selective participation biases in genetic or non-genetic analyses.

## Results

**Observational associations.** The demographics of the participants included in this study are summarised (Table 1). Overall, 42,429 participants completed all four optional components of the UK Biobank study, whilst 51,141 participated in the food frequency questionnaire (FFQ), the physical activity actigraph monitoring and the mental health questionnaire (MHQ).

Participation in the four additional UK Biobank questionnaires and tests was associated with older age (FFQ and aide-memoire), female sex (all four outcomes), lower body mass index (all four outcomes), lower levels of deprivation (all four outcomes), higher fluid intelligence (all four outcomes), never smoking (all four outcomes), higher self-reported physical activity using the International Physical Activity Questionnaire (IPAQ) (FFQ and physical activity), higher measured physical activity (aide-memoire, MHQ), no depression (MHQ and aide-memoire) and no type 2 diabetes (all four outcomes). There was some evidence that the aide-memoire variable captured a different aspect of participation, with associations in the opposite direction to the other participation measures. For example, a longer duration in education was associated with lower odds of completing the aide-memoire, but higher odds of participating in the other three components. This was further evidenced by strong observational associations and genetic correlations between three of the participation variables, whilst completing the aide-memoire was not as robustly correlated with participation in the other optional surveys (Supplementary Tables 1 and 2).

We also generated a binary variable to compare participants who were invited to participate in at least one the optional surveys (i.e., FFQ, MHQ and physical activity) versus those participants not invited (Supplementary Table 3). Receiving an invite to participate in the three optional surveys ($n = 336,633$) was associated with younger age, male sex, lower body mass index, lower levels of deprivation, higher fluid intelligence, never smoking and a lower prevalence of type 2 diabetes.

**GWAS of the participation variables identified 32 loci.** GWAS of the four participation traits was performed in individuals of European descent using BOLT-LMM, with sample sizes of $N = 300,639$ for the FFQ, $N = 215,127$ for physical activity monitoring, $N = 294,787$ for MHQ and $N = 451,306$ for aide-memoire. After clumping and using a stringent GWAS cut-off of $P < 6 \times 10^{-9}$, there were 8 loci for the FFQ, 1 locus for physical activity participation, 21 loci for MHQ participation and 2 loci for aide-memoire (Table 2 and Supplementary Fig. 1). Twenty-three variants were associated at $P < 6 \times 10^{-9}$ with receiving an invite to participate in any of the optional surveys (Supplementary Data 1). All variants reaching the less stringent $P < 5 \times 10^{-8}$ threshold are reported (Supplementary Tables 4 and 5). With the exception of the aide-memoire, many of the lead variants for the other participation measures were within 500 kb of another lead variant for a different participation measure (Table 2 and Supplementary Data 2). For example, 6/8 of the FFQ lead variants at $P < 6 \times 10^{-9}$ were within 500 kb of another lead variant for either actigraphy or MHQ participation, whilst the only variant at $P < 6 \times 10^{-9}$ for actigraphy was within 500 kb of an FFQ and MHQ variant and 4/21 variants at $P < 6 \times 10^{-9}$ for the MHQ were within 500 kb of an FFQ or actigraphy variant.

Two of the variants identified for FFQ participation were previously identified in GWAS of intelligence (rs11210871[11,12])

**Table 1 Demographics of the four participation measures.**

| Demographic | Summary demographics for those with food frequency data | Odds (95% CI) of participation in FFQ per unit change in demographic** | P* |
|---|---|---|---|
| N | 300,639 | | |
| Mean age at baseline (SD) | 56.5 (8.0) | 1.06 (1.06, 1.07) | $<1\times10^{-15}$ |
| Male sex, N (%) | 139,941 (46.6) | 0.87 (0.85, 0.88) | $<1\times10^{-15}$ |
| Mean BMI at baseline (SD) | 27.2 (4.7) | 0.83 (0.83, 0.84) | $<1\times10^{-15}$ |
| Mean Townsend deprivation index (SD) | −1.73 (2.8) | 0.99 (0.98, 1.00) | 0.006 |
| Mean years in education (SD)*** | 15.8 (4.7) | 1.06 (1.06, 1.07) | $<1\times10^{-15}$ |
| Mean systolic blood pressure (SD) | 143 (23) | 0.96 (0.95, 0.97) | $<1\times10^{-15}$ |
| Mean fluid intelligence (SD) | 6.1 (2.1) | 1.19 (1.18, 1.20) | $<1\times10^{-15}$ |
| Mean self-reported physical activity | 7.4 (1.1) | 0.98 (0.98, 0.99) | $5\times10^{-6}$ |
| Mean physical activity (accelerometer)**** | 0.21 (0.07) | 0.97 (0.96, 0.98) | $8\times10^{-8}$ |
| Smoking status, N (%) | | | |
| Never | 165,312 (55.0) | Reference | |
| Former | 108,888 (36.2) | 0.94 (0.92, 0.95) | $<1\times10^{-15}$ |
| Current | 22,862 (7.6) | 0.68 (0.67, 0.70) | |
| Type 2 diabetes, N (%) | 12,730 (4.2) | 0.83 (0.80, 0.86) | $<1\times10^{-15}$ |
| Depression, N (%) | 36,423 (12.1) | 1.00 (0.98, 1.02) | 0.96 |

| Demographic | Yes | No | Odds (95% CI) of participating in physical activity monitoring per unit change in demographic | P* |
|---|---|---|---|---|
| N | 96,035 | 119,092 | | |
| Mean age at baseline (SD) | 56.7 (7.8) | 56.7 (8.0) | 1.00 (0.99, 1.01) | 0.59 |
| Male sex, N (%) | 42,226 (44.0) | 56,273 (47.3) | 0.88 (0.86, 0.89) | $<1\times10^{-15}$ |
| Mean BMI at baseline (SD) | 26.7 (4.5) | 27.4 (4.7) | 0.86 (0.85, 0.87) | $<1\times10^{-15}$ |
| Mean Townsend deprivation index (SD) | −1.80 (2.8) | −1.75 (2.8) | 0.98 (0.97, 0.99) | $5\times10^{-5}$ |
| Mean years in education (SD)*** | 16.2 (4.6) | 15.4 (4.9) | 1.04 (1.03, 1.04) | $<1\times10^{-15}$ |
| Mean systolic blood pressure (SD) | 142 (23) | 143 (24) | 0.92 (0.92, 0.93) | $<1\times10^{-15}$ |
| Mean fluid intelligence (SD) | 6.04 (2.11) | 5.98 (2.08) | 1.04 (1.03, 1.05) | $1\times10^{-12}$ |
| Mean self-reported physical activity | 7.42 (1.08) | 7.36 (1.13) | 1.05 (1.04, 1.06) | $<1\times10^{-15}$ |
| Mean physical activity (accelerometer)**** | 0.21 (0.07) | NA | NA | NA |
| Smoking status, N (%) | | | | |

**Table 1 (continued)**

| Demographic | Yes | No | Odds (95% CI) of participating in physical activity monitoring per unit change in demographic | P* |
|---|---|---|---|---|
| Never | 54,352 (56.6) | 64,944 (54.5) | Reference | <1 × 10⁻¹⁵ |
| Former | 34,818 (36.3) | 42,781 (35.9) | 0.98 (0.97, 1.00) | |
| Current | 5860 (6.1) | 9857 (8.3) | 0.72 (0.70, 0.74) | |
| Type 2 diabetes, N (%) | 3260 (3.4) | 5428 (4.6) | 0.76 (0.73, 0.80) | <1 × 10⁻¹⁵ |
| Depression, N (%) | 11,791 (12.3) | 13,528 (11.4) | 1.04 (1.02, 1.07) | 0.002 |

| Demographic | Yes | No | Odds (95% CI) of participating in mental health questionnaire per unit higher demographic variable for continuous | P* |
|---|---|---|---|---|
| N | 146,074 | 148,713 | | |
| Mean age at baseline (SD) | 56.6 (7.7) | 56.5 (8.2) | 1.00 (1.00, 1.01) | 0.26 |
| Male sex, N (%) | 63,586 (43.5) | 72,063 (48.5) | 0.82 (0.81, 0.83) | <1 × 10⁻¹⁵ |
| Mean BMI at baseline (SD) | 26.8 (4.6) | 27.5 (4.7) | 0.86 (0.85, 0.87) | <1 × 10⁻¹⁵ |
| Mean Townsend deprivation index (SD) | −1.79 (2.8) | −1.70 (2.8) | 0.97 (0.96, 0.97) | <1 × 10⁻¹⁵ |
| Mean years in education (SD)*** | 16.4 (4.5) | 15.3 (4.9) | 1.06 (1.05, 1.06) | <1 × 10⁻¹⁵ |
| Mean systolic blood pressure (SD) | 141 (23) | 143 (24) | 0.91 (0.90, 0.91) | <1 × 10⁻¹⁵ |
| Mean fluid intelligence (SD) | 6.1 (2.1) | 6.0 (2.1) | 1.10 (1.09, 1.12) | <1 × 10⁻¹⁵ |
| Mean self-reported physical activity | 7.4 (1.1) | 7.4 (1.1) | 0.99 (0.98, 1.00) | 0.05 |
| Mean physical activity (accelerometer)**** | 0.21 (0.07) | 0.20 (0.07) | 1.03 (1.01, 1.04) | 0.0008 |
| Smoking status, N (%) | | | | |
| Never | 83,571 (57.2) | 79,862 (53.7) | Reference | <1 × 10⁻¹⁵ |
| Former | 51,859 (35.5) | 54,247 (36.5) | 0.92 (0.91, 0.94) | |
| Current | 9062 (6.2) | 9062 (6.2) | 0.69 (0.67, 0.71) | |
| Type 2 diabetes, N (%) | 4692 (3.2) | 7311 (4.9) | 0.69 (0.66, 0.71) | <1 × 10⁻¹⁵ |
| Depression, N (%) | 17,660 (12.1) | 18,002 (12.1) | 0.95 (0.93, 0.98) | 6 × 10⁻⁵ |

| Demographic | Yes | No | Odds (95% CI) of completing aide-memoire per unit change in demographic | P* |
|---|---|---|---|---|
| N | 361,501 | 89,535 | | |
| Mean age at baseline (SD) | 57.6 (7.9) | 55.9 (8.1) | 1.24 (1.23, 1.25) | <1 × 10⁻¹⁵ |
| Male sex, N (%) | 162,332 (44.9) | 43,887 (49.0) | 0.84 (0.83, 0.85) | <1 × 10⁻¹⁵ |
| Mean BMI at baseline (SD) | 27.4 (4.8) | 27.6 (4.8) | 0.95 (0.94, 0.96) | <1 × 10⁻¹⁵ |
| Mean Townsend deprivation index (SD) | −1.55 (2.9) | −1.17 (3.1) | 0.90 (0.90, 0.91) | <1 × 10⁻¹⁵ |
| Mean years in education (SD)*** | 14.8 (5.1) | 15.2 (5.1) | 0.99 (0.99, 0.99) | <1 × 10⁻¹⁵ |
| | 145 (24.2) | 142 (23) | 1.07 (1.06, 1.08) | <1 × 10⁻¹⁵ |

**Table 1 (continued)**

| Demographic | Yes | No | Odds (95% CI) of completing aide-memoire per unit change in demographic | P* |
|---|---|---|---|---|
| Mean systolic blood pressure (SD) | 5.9 (2.1) | 5.9 (2.1) | 1.03 (1.01, 1.04) | $3 \times 10^{-6}$ |
| Mean fluid intelligence (SD) | 7.4 (1.1) | 7.4 (1.1) | 1.00 (0.99, 1.00) | 0.98 |
| Mean self-reported physical activity | | | | |
| Mean physical activity (accelerometer)**** | 0.21 (0.07) | 0.20 (0.07) | 1.05 (1.03, 1.07) | $9 \times 10^{-9}$ |
| Smoking status, N (%) | | | | |
| Never | 195,889 (54.2) | 46,781 (52.3) | Reference | $<1 \times 10^{-15}$ |
| Former | 128,821 (35.6) | 31,082 (34.7) | 0.94 (0.93, 0.96) | |
| Current | 32,165 (8.9) | 10,209 (11.4) | 0.79 (0.77, 0.81) | |
| Type 2 diabetes, N (%) | 18,130 (5.0) | 4425 (4.9) | 0.94 (0.91, 0.97) | 0.0003 |
| Depression, N (%) | 38,535 (10.7) | 10,541 (11.8) | 0.88 (0.86, 0.90) | $<1 \times 10^{-15}$ |

*Age and sex-adjusted logistic regression models.
**Age and sex-adjusted ordinal regression models.
***Not inverse normalised so odds per unit change.
****Activity proportion over 40 mg.

and cognitive performance (rs13428598[13]). For both variants, the allele associated with higher intelligence or cognitive performance is associated with completing more FFQ. A further two variants (rs9261655 and rs147412694) were associated with blood cell traits[14]. Here, the alleles associated with higher blood cell counts were associated with completing fewer FFQ, whilst the reason behind this unknown, higher white blood cell counts have previously been associated with poorer cognition[15,16]. Four of the eight variants were also in high LD ($r^2 > 0.8$) with genome-wide significant (GWS) signals from behavioural GWAS, including ADHD, risk tolerance, smoking and alcohol consumption. As expected from previous work on participation and our understanding of risky behaviours, alleles associated with a higher risk of ADHD, higher-risk tolerance and a higher risk of smoking or consuming alcohol associated with lower FFQ participation (Table 2).

The locus identified for participation in actigraphy (rs55714359) was in partial linkage disequilibrium with the variants identified for participation in the mental health questionnaire ($r^2 = 0.52$) and completing the food frequency questionnaire ($r^2 = 0.32$). This variant was previously identified as associated with multiple sclerosis[17], with the allele associated with higher odds of multiple sclerosis associated with lower odds of participation in physical activity. In the UK Biobank, this variant is also associated with adiposity related traits[18]. The allele associated with higher adiposity is also associated with lower odds of participation in physical activity monitoring.

Of the 25 loci identified in a GWAS of MHQ participation by Adams et al.[6] we replicated 15/25 (60%) at $P < 6 \times 10^{-9}$ and 22/25 (88%) at $P < 5 \times 10^{-8}$ in this larger sample of related individuals. The three missing variants (rs35028061, rs13082026 and rs57692580) were directionally consistent and approaching GWS (P values were $5.1 \times 10^{-8}$, $6.6 \times 10^{-7}$ and $9.6 \times 10^{-7}$, respectively). Of the 21 variants associated with MHQ participation at the stringent threshold, four were previously associated with cognitive function and intelligence measures (rs7542974, rs485929, rs11793831 and rs7108020[13]) and a further three were in high LD with variants identified associated with intelligence outcomes. For all variants, the allele associated with higher intelligence or cognitive performance was associated with higher odds of completing the MHQ.

A missense mutation in *APOE* (rs429358) was associated with MHQ participation. The C-allele is a marker of the APOE-ε4 genotype which is a major risk factor for Alzheimer's disease[19], and here, was associated with lower odds of participation in the MHQ. Further analysis in the unrelated subset tested whether individuals with APOE-ε4ε4 haplotype were less likely to participate in the MHQ compared to those with the APOE-ε2ε2 haplotype. Lower odds of MHQ participation was observed in the APOE-ε4ε4 haplotype carriers 0.89 (95% CI: 0.80, 1.00) in all individuals and in those who were less than 50 years old at recruitment (OR: 0.81 (95% CI: 0.65, 1.00)). This suggests that individuals with early signs of cognitive impairment had reduced capacity to participate in the MHQ.

The variant rs58101275 has previously been associated with bone mineral density[20] and isoleucine levels[21]. The G allele raises both isoleucine levels and bone mineral density (BMD) and was associated with lower odds of completing the aide-memoire. Previous studies have demonstrated that BMD is inversely associated with cognition[22] and Alzheimer's disease, indicating those with higher BMD may have a better memory.

Of the 23 variants at $P < 6 \times 10^{-9}$, 6 were either top signals for MHQ participation or in high LD ($r^2 > 0.8$) with variants for MHQ participation (Supplementary Table 3). For all six loci, the allele that was associated with higher odds of participation in the MHQ was associated with higher odds of receiving an invite to

**Table 2 Variants associated with participation from genome-wide association analyses in the UK Biobank ($P < 6 \times 10^{-9}$).**

| Participation measure | SNP | Chromosome | Location (Bp) | A1/A2 | Frequency | OR (95% CI) or beta (SE) | P value* | Hit within 500 KB of another participation measure? | Identified in previous GWAS | Known association signal | GWAS hits for loci in high LD ($r^2 > 0.8$) |
|---|---|---|---|---|---|---|---|---|---|---|---|
| FFQ | rs11210871 | 1 | 44,029,353 | C/G | 0.30 | −0.59 (0.10) | 1.80E-09 | Yes (MHQ) | NA | Associated with intelligence—C-allele lower intelligence (PMIDs: 29326435, 29942086) | In LD with variants associated with ADHD (rs11210887, $r^2 = 0.86$, PMID: 30610198), smoking initiation (rs3001723, $r^2 = 0.90$, PMID: 30617275), risk tolerance (rs3001723, $r^2 = 0.91$, PMID: 30643258) and schizophrenia (rs3001723, $r^2 = 0.91$, PMID: 26198765) |
| FFQ | rs76473275 | 1 | 243,460,555 | T/C | 0.85 | −0.88 (0.13) | 3.10E-12 | | NA | Associated with cognitive function, intelligence and educational attainment (PMIDs: 29326435, 30038396) | In LD with variants associated with chronotype (rs28380327, $r^2 = 0.94$, PMID: 30696823) and neuroticism (rs10048736, $r^2 = 0.94$, PMID: 29942085) |
| FFQ | rs13428598 | 2 | 144,250,487 | C/T | 0.61 | −0.65 (0.09) | 1.20E-12 | Yes (MHQ) | NA | | |
| FFQ | rs11134465 | 5 | 167,037,934 | G/A | 0.32 | −0.60 (0.10) | 4.40E-10 | Yes (MHQ) | NA | Associated with blood cell traits (PMID: 27863252) | |
| FFQ | rs9261655 | 6 | 30,288,283 | G/C | 0.89 | 0.86 (0.15) | 4.00E-09 | | NA | | |
| FFQ | rs2622102 | 7 | 153,495,423 | A/G | 0.52 | 0.60 (0.09) | 5.30E-11 | Yes (MHQ) | NA | | In LD with variants associated with alcohol consumption (rs6951574, $r^2 = 0.91$, PMID: 30643251), regular attendance at pub or social club (rs6969458, $r^2 = 0.92$, PMID: 29970889), smoking status (rs6951574, $r^2 = 0.91$, PMID: 30595370), risk-taking (rs2533148, $r^2 = 0.86$, PMID: 30643258) |
| FFQ | rs200373 | 19 | 18,286,546 | T/A | 0.51 | 0.52 (0.09) | 4.80E-09 | Yes (physical activity and MHQ) | NA | | |
| FFQ | rs147412694 | 21 | 40,702,786 | G/A | 0.85 | −0.77 (0.13) | 9.60E-10 | | NA | Associated with blood cell traits (PMID: 27863252) | In LD with variants associated with smoking status (rs77217252, $r^2 = 0.85$, PMID: 30595370) |
| Physical activity | rs55714539 | 19 | 18,207,397 | A/C | 0.66 | 1.08 (1.05, 1.11) | 1.30E-09 | Yes (FFQ and MHQ) | NA | Associated with multiple sclerosis (PMID: 24076602) | |
| MHQ | rs7542974 | 1 | 72,544,704 | G/A | 0.75 | 0.97 (0.96, 0.98) | 1.30E-09 | | Yes, lead SNP | Associated with cognitive function (PMID: 30038396) and intelligence (PMID: 29942086) | |
| MHQ | rs485929 | 1 | 74,678,285 | A/G | 0.61 | 0.97 (0.96, 0.98) | 4.00E-10 | | Yes, lead SNP | Associated with cognitive function (PMID: 30038396) and intelligence (PMID: 29942086) | |
| MHQ | rs618232 | 1 | 84,357,225 | A/C | 0.26 | 1.03 (1.02, 1.04) | 1.30E-09 | | Yes, within 500 kb of SNP rs532246 ($r^2 = 0.9947$) | | |
| MHQ | rs1565440 | 1 | 243,387,788 | G/A | 0.63 | 1.03 (1.02, 1.04) | 6.50E-13 | Yes (FFQ) | Yes, within 500 kb of SNP rs2789111 ($r^2 = 0.8612$) | Associated with risk-taking behaviour (PMID: 30271922) | |
| MHQ | rs1012940 | 2 | 184,534,996 | A/C | 0.70 | 0.97 (0.96, 0.98) | 9.70E-10 | | No | | |
| MHQ | rs34631 | 5 | 60,526,326 | T/C | 0.52 | 0.97 (0.96, 0.98) | 1.20E-10 | | Yes, within 500 kb of rs34635 ($r^2 = 0.6629$) | Associated with cognitive function (PMID: 30038396) and intelligence (PMID: 29942086) | In LD with variants associated with self-reported maths ability (rs194369, $r^2 = 0.98$, PMID: 30038396) and general cognitive ability (rs34627, $r^2 = 1$, PMID: 29844566) |
| MHQ | rs2844472 | 6 | 31,589,676 | A/G | 0.65 | 1.03 (1.02, 1.04) | 1.60E-09 | Yes (FFQ) | Yes, within 500 kb of rs3993747 ($r^2 = 0.9636$) | Associated with multiple sclerosis (PMID: 24076602, rheumatoid arthritis) (PMID: 23143596), blood traits (PMID: 27863252), psoriasis (PMID: 23143594) and SLE (PMID: 26502338) | In LD with a variant associated with height (rs2857693, $r^2 = 0.89$, PMID: 25282103) |
| MHQ | rs11793831 | 9 | 23,362,311 | G/T | 0.58 | 0.97 (0.96, 0.98) | 8.60E-11 | | Yes, lead SNP | Associated with cognitive performance (PMID: 30038396), intelligence (PMID: 29942086), self-reported educational attainment (PMID: 27046643) and bipolar disorder (PMID: 27329760) | In LD with a variant associated with educational attainment (rs3897821, $r^2 = 0.86$, PMID: 30038396) |
| MHQ | rs2236295 | 10 | 64,564,892 | G/T | 0.60 | 1.03 (1.02, 1.04) | 5.90E-09 | | No | Associated with urinary albumin—G allele higher albumin (PMID: 30220432) | |
| MHQ | rs7910869 | 10 | 67,964,514 | T/C | 0.79 | 0.97 (0.96, 0.98) | 7.60E-10 | | No | | |
| MHQ | rs1223114 | 11 | 31,523,130 | A/G | 0.39 | 1.03 (1.02, 1.04) | 1.80E-09 | | Yes, within 500 kb of rs1984389 ($r^2 = 0.6852$) | | In LD with a variant associated with diastolic blood pressure (rs10995311, $r^2 = 0.81$, PMID: 27618447) |

**Table 2 (continued)**

| Participation measure | SNP | Chromosome | Location (Bp) | A1/A2 | Frequency | OR (95% CI) or beta (SE) | P value* | Hit within 500 KB of another participation measure? | Identified in previous GWAS | Known association signal | GWAS hits for loci in high LD ($r^2$ >0.8) |
|---|---|---|---|---|---|---|---|---|---|---|---|
| MHQ | rs17145219 | 11 | 83,167,337 | C/G | 0.89 | 1.04 (1.03, 1.06) | 2.90E-09 | | No | | |
| MHQ | rs7108020 | 11 | 131,289,820 | C/A | 0.36 | 0.97 (0.96, 0.98) | 8.70E-12 | | Yes, within 500 kb of rs1079143 ($r^2$ = 0.8252) | Associated with educational attainment and highest math class taken—A associated with higher attainment (PMID: 30038396) | |
| MHQ | rs35917376 | 15 | 75,595,357 | T/A | 0.25 | 1.03 (1.02, 1.04) | 4.40E-09 | Yes (FFQ) | No | | |
| MHQ | rs8055041 | 16 | 7,462,715 | C/G | 0.52 | 0.97 (0.97, 0.98) | 1.40E-09 | | Yes, within 500 kb of rs4616299 ($r^2$ = 0.7553) | | |
| MHQ | rs7207531 | 17 | 56,426,789 | G/A | 0.58 | 0.97 (0.96, 0.98) | 1.30E-09 | | Yes, within 500 kb of rs56058331 ($r^2$ = 1) | | |
| MHQ | rs1261078 | 18 | 52,866,791 | A/G | 0.95 | 1.07 (1.05, 1.09) | 5.50E-12 | | Yes, lead SNP | Associated with haematological traits (PMID: 27863252) | |
| MHQ | rs35502362 | 19 | 4,966,041 | C/T | 0.65 | 0.97 (0.96, 0.98) | 3.20E-09 | | Yes, within 500 kb of rs34232444 ($r^2$ = 1) | | |
| MHQ | rs3746187 | 19 | 18,279,816 | A/G | 0.60 | 1.03 (1.02, 1.04) | 1.80E-10 | Yes (FFQ and actigraphy) | Yes, lead SNP | | In LD with a variant associated with the number of sexual partners (rs273512, $r^2$ = 0.84, PMID: 30643258) |
| MHQ | rs429358 | 19 | 45,411,941 | T/C | 0.85 | 1.06 (1.05, 1.07) | 1.10E-20 | | Yes, lead SNP | Highly pleiotropic APOE variant—associated with Alzheimer's disease, cholesterol, CRP etc. | |
| MHQ | rs1232482 | 20 | 11,886,643 | C/T | 0.60 | 0.97 (0.96, 0.98) | 2.30E-09 | | No | | In LD with variants associated with diastolic blood pressure (rs1232482, $r^2$ = 1, PMID: 30224653, cigarettes per day (rs6078373, $r^2$ = 0.93, PMID: 30643251) |
| Aide-memoire | rs2049604 | 7 | 113,990,352 | C/T | 0.64 | 0.97 (0.96, 0.98) | 4.60E-09 | | NA | | |
| Aide-memoire | rs58101275 | 14 | 104,008,420 | G/A | 0.79 | 0.96 (0.95, 0.98) | 5.00E-09 | | NA | Associated with heel bone mineral density (PMID: 30598549) and isoleucine levels (PMID: 27898682) | |

*P value from BOLT-LMM GWAS analyses.

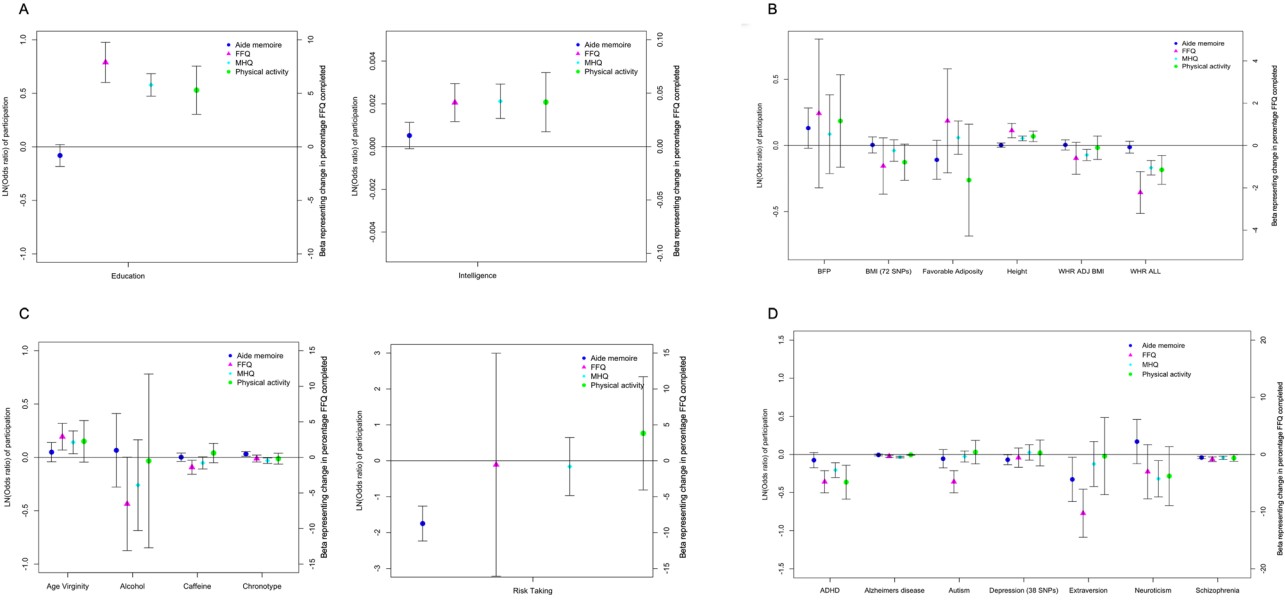

**Fig. 1 Plots of the Mendelian Randomisation results.** Dot plots representing the inverse-variance weighted results from two-sample MR analyses for (**A**) educational, (**B**) anthropometric, (**C**) behavioural and (**D**) neurological and psychological traits. Error bars represent the 95% confidence intervals of the IVW estimate.

participate in at least one optional survey. Variant rs73078357 was previously identified as associated with email contact (Supplementary Table 3). 8/23 variants were previously associated with cognitive performance[13], intelligence[12] and self-reported educational attainment[13,23].

**Genetic correlations with published GWAS studies**. After Bonferroni correction ($P < 1.5 \times 10^{-5}$), we observed strong positive genetic correlations between three of the four participation measures (FFQ, MHQ and physical activity completion) and qualifications, fluid intelligence, years spent in education. Strong inverse genetic correlations were noted between three of the four participation measures (FFQ completion, MHQ and physical activity completion) and obesity-related traits. Completing the aide-memoire, was strongly inversely correlated with risk-taking, ever smoking, driving fast, having fractured bones in the last 5 years and schizophrenia. It was positively associated with suffering from nerves and experiencing nervous feelings.

**Genetic correlations with ALSPAC participation measures**. There were positive genetic correlations between the ALSPAC participation measures and FFQ completion (mother: rg = 0.533, $P = 3 \times 10^{-8}$; child: rg = 0.488, $P = 3 \times 10^{-9}$), participation in MHQ (mother: rg = 0.616, $P = 8 \times 10^{-10}$, child: rg = 0.627, $P = 2 \times 10^{-12}$), and physical activity participation (mother rg = 0.487, $P = 2 \times 10^{-5}$; child rg = 0.319, $P = 0.001$) (Supplementary Table 2). The aide-memoire variable in UK Biobank was not strongly correlated with the ALSPAC participation measures (mother rg = 0.215, $P = 0.08$, child rg = 0.167, $P = 0.14$; Supplementary Table 2). Receiving an invite to participate was strongly correlated with the participation measures in ALSPAC (mother rg = 0.58, $P = 1 \times 10^{-9}$; child rg = 0.59, $P = 1 \times 10^{-10}$).

**Mendelian randomisation analyses**. In all individuals, Mendelian randomisation[24,25] analysis demonstrated that 27 traits caused at least one participation measure at a threshold of $P < 0.05$ ($P$ value based on the inverse-variance weighted (IVW) analyse, with 8 at more stringent $P < 0.0001$; Supplementary

Table 6). Of the 27 traits, 15, 18, 10 and 6 were associated with FFQ, MHQ and physical activity and aide-memoire, respectively.

Longer duration in education and higher intelligence predicted higher odds of participation in the FFQ, MHQ and physical activity monitoring (Fig. 1A and Supplementary Table 6). For example, a one-SD longer duration (~5 years) in education caused higher odds of participation in the MHQ (1.78 (95% CI: 1.61, 1.98)) and physical activity monitoring (1.69 (95% CI: 1.36, 2.13)). In contrast, there was limited evidence for longer educational duration predicting the completion of the aide-memoire.

Higher adiposity caused lower odds of participation in the FFQ, MHQ and physical activity monitoring. For example, the odds ratios for participation in the MHQ and PA monitoring per one-SD higher waist:hip ratio were 0.85 (95% CI: 0.80, 0.89) and 0.83 (95% CI: 0.75, 0.93), respectively (Fig. 1B and Supplementary Data 3). Higher BMI caused lower odds of participation in the FFQ and physical activity monitoring in women (Supplementary Data 3). There was limited evidence that higher adiposity predicted aide-memoire completion. A one-SD taller stature caused higher odds of completing the MHQ (OR: 1.06 (95% CI: 1.04, 1.07)) and physical activity monitoring (OR: 1.07 (95% CI: 1.03, 1.11)). Taller stature also caused participants to complete more FFQ (Fig. 1B and Supplementary Data 3). There was no strong evidence that any of the other anthropometric measures tested caused participation, although many of the estimates have wide confidence intervals.

Genetic evidence demonstrated that behavioural characteristics caused participation (Fig. 1C). For example, older age of losing virginity caused participants to complete more FFQ and have higher odds of participation in the MHQ (OR: 1.15 (95% CI: 1.03, 1.28)). A twofold higher genetic liability for being a morning person chronotype caused higher odds of completing the aide-memoire (OR: 1.02 (95% CI: 1.01, 1.04)) and lower odds of completing the MHQ (OR: 0.98 (95% CI: 0.96, 1.00)). A twofold higher genetic liability for riskier behaviour caused lower odds of completing the aide-memoire (OR: 0.27 (95% CI: 0.19, 0.40)), but was not linked to completing the optional surveys. In a subset of former and current smokers, the role of smoking heaviness on participation was explored. A one-SD higher cigarette per day

(~11 cigarettes) caused lower odds of participating in the MHQ (OR: 0.88 (95% CI: 0.85, 0.92)) and the physical activity monitoring (OR: 0.93 (95% CI: 0.89, 0.97)) (Supplementary Data 4).

C-reactive protein (CRP) was the only biomarker tested with some evidence of a causal association, with a twofold higher CRP causing higher odds of completing the MHQ (OR: 1.08 (95% CI: 1.04, 1.12); Supplementary Data 3).

Higher genetic liability of cancer and non-cancer diseases and poorer metabolic health generally caused lower odds of participation (Supplementary Data 3). For example, a twofold higher genetic liability of breast cancer was associated with lower odds of participating in the MHQ (OR: 0.98 (95% CI: 0.96, 1.00)), physical activity monitoring (OR: 0.97 (95% CI: 0.95, 1.00)) and completing the aide-memoire (OR: 0.97 (95% CI: 0.95, 1.00)).

Several psychological and neurological conditions caused lower odds of participation (Fig. 1D and Supplementary Data 4). For example, a genetic liability to ADHD and schizophrenia was associated with the completion of fewer FFQ and lower odds of participation in the MHQ and physical activity monitoring. A twofold higher genetic risk of schizophrenia lowered the odds of completing the MHQ by 3%, (OR: 0.97 (95% CI: 0.95, 0.99)). A genetic liability for schizophrenia also lowered the odds of completing the aide-memoire. Genetic liability for autism and extraversion caused fewer FFQ to be completed. Alzheimer's disease genetic liability was associated with lower odds of participation in the FFQ, physical activity monitoring and MHQ. A doubling in Alzheimer's genetic liability was associated with a 0.976 (95% CI: 0.969, 0.983) lower odds of completing the MHQ.

There was little evidence that reproductive traits in women caused participation, with the exception of age at menarche. For example, a one year older age at menarche was associated with 1.07 (95% CI: 1.03, 1.11) and 1.07 (95% CI: 1.03, 1.12) higher odds of completing the MHQ and physical activity monitoring, respectively (Supplementary Data 3).

Generally, results were consistent when analysed in men and women separately (Supplementary Data 3), with the exception of BMI and physical activity participation, where evidence suggested high BMI only caused lower odds of participation in women ($OR_{women}$: 0.88 (95% CI: 0.81, 0.96), $OR_{men}$: 1.01 (95% CI: 0.92, 1.12)), $P_{interaction} = 0.07$).

Two-sample MR methods that are more robust to pleiotropy generally provided similar results (Supplementary Data 3).

## Discussion

This study explored the genetic basis of four different participation measures, plus whether or not participants were invited to at least one optional element in the UK Biobank study and used Mendelian randomisation to test the causal role of a broad range of factors in participation.

Some individual characteristics appear to decrease the likelihood of participation in all of the optional invited components of the UK Biobank study (i.e., physical activity monitoring, food frequency and MHQ). These include lower intelligence and educational attainment, higher adiposity and increased liability to ADHD and neuroticism and schizophrenia. Many of these were previously identified in the ALSPAC study[1,5], previous UK Biobank study analyses[6] and Generation Scotland[6]. This implies that missingness of all the variables collected in the optional components of UK Biobank will be influenced by these underlying traits. The fourth participation measure considered was the aide-memoire, where participants were asked at baseline to complete a short form with specific data. Our analyses suggest that this

measure captures another aspect of behaviour, perhaps reflecting compliance rather than participation, with evidence that a genetic liability to riskier behaviour was inversely associated with completing the aide-memoire.

GWAS identified a number of loci robustly associated with the different participation measures. A number of genome-wide significant loci were shared across the participation traits, suggesting a general role in influencing participation. Although further analyses using colocalization methods would be necessary to more formally test whether these shared loci represent the same signal. Many of the variants identified were in or near loci which had previously been identified as associating with, intelligence and cognitive function or behaviour-based traits. Alleles associated with higher intelligence or risk aversion were consistently associated with completing the MHQ and more of the FFQ. In the MHQ GWAS, the top signal was in the highly pleiotropic APOE locus. The allele that raises participation in the MHQ (T) is associated with lower odds of Alzheimer's disease[19], heart disease, inflammation and dyslipidaemia[26]. Further analyses indicated that the APOE-ε4ε4 haplotype carriers were less likely to participate in the MHQ and high genetic liability for Alzheimer's disease lowered odds of participation in the FFQ, physical activity monitoring and the MHQ.

In addition to performing GWAS of our four participation measures, we also performed a GWAS of invitation to participate in at least one of the three optional components. Because only those invited can participate, the fact that not everyone is invited could result in collider bias in our analysis of participation[27]. A factor that is positively associated with both being invited and participation is likely to have its association with participation biased towards the null when conditioning on having been invited, assuming that being invited and participating are positively correlated (as demonstrated here) and that there are no interactions (on the probability scale) between the variable and others that also influence invitation/participation. Here, we demonstrated that some variants were associated with higher odds of both being invited to participate and completing the MHQ. This suggests that here conditioning on being invited to participate could have resulted in the Mendelian randomisation analyses for these variables being biased towards the null, if they were in truth positively associated with participation. However, if there are non-linearities or interactions in the effects of the risk factors on invitation/participation, then the direction of the bias cannot be predicted. Similarly, a factor that affects being invited, but does not in truth affect participation, could appear to have a positive or negative spurious association with participation, conditional on being invited.

Using genetic correlation analyses, we have demonstrated that these participation issues are not specific to the UK Biobank. Two participation measures from the ALSPAC study[1] were strongly correlated with the participation measures derived in the UK Biobank. This fits with a previous study where strong genetic correlations were noted between UK Biobank mental health participation and participation in follow-up in Generation Scotland[6]. These results suggest that similar genetic factors are driving participation in follow-up and optional components of studies, regardless of study design, recruitment strategies and the population demographics of the study. The similarity of factors affecting participation across different studies is potentially important for comparisons of results between studies—if similar factors cause participation in different studies, then collider bias will have the same impact on the results from each study. Thus, results from different studies would be subject to similar biases, causing replication of results across studies to become meaningless.

These results are important for informing analysis strategies and the likely direction and magnitude of bias due to conditioning only on those who participate. For the participator-only analysis for a given model to be unbiased, it is necessary for the outcome variable to be independent of missingness, given the variables in the analysis model[7]. Thus when examining the factors affecting physical activity, all the factors that we have shown here to be related to participation in the physical activity monitoring (BMI, height, education, intelligence, ADHD, age at menarche), should be either included in the analysis model or used in other strategies such as inverse probability weighting (IPW) or multiple imputations (MI). Where a selection is related to the underlying concept(s) measured by the optional component, then this concept will be missing not at random and analyses where it is the outcome will likely be biased[7]. On the other hand, a participator-only analysis of a model that involves only characteristics that are unrelated to participation will not be biased by conditioning on participation.

Selection of the type demonstrated here may cause bias in estimates of effects, and the size and direction of bias cannot (usually) be exactly determined. Previous work has shown that estimated effects of risk factors on mortality and cause-specific mortality differ between UK Biobank and the less-selected Health Survey for England—with some being moved towards, and some away from the null. This could imply that selection into UK Biobank is causing bias in estimating these effects. For example, we have shown that smoking is negatively associated with participation in UK Biobank. If a factor that causes lung cancer is also negatively associated with participation (e.g., socioeconomic position), then selecting on participating in UK Biobank would induce a negative association between smoking and lung cancer (assuming an additive model), which would bias the estimated effect of smoking on lung cancer towards the null. This is indeed what is seen in the comparison of estimates for this effect between UK Biobank and HSE-SHS[9]. It should be noted that this simple estimate of the direction of bias depends on assumptions about the underlying selection model, and cannot be verified with only UK Biobank data—e.g., an interaction between smoking and socioeconomic position in their effect on participation could change the size and direction of any bias. We have similarly shown previously that some effect estimates were different when calculated on only those continuing to participate in ALSPAC, compared to all those participating at baseline[1]. It has also been suggested that selection bias may (at least in part) be responsible for overestimates of the protective effect of moderate alcohol consumption[28,29].

Strategies to investigate or minimise the impact of selection on a given estimate depend on the data available on the population not selected into the study. Inverse probability weighting (IPW) has been suggested to overcome mortality bias[30], but the validity of this depends on correctly specifying the selection model. If there is an unmeasured factor that affects selection and is related to the variables in the analysis model, then this may mean that inverse probability weighting is not unbiased[31]. IPW as a solution also depends on having data on all the variables affecting selection and their distribution in the population in which we wish to make the inference. Solutions using IPW to infer bounds on estimates have been proposed, although these can result in wide bounds, or depend on underlying assumptions about associations of unmeasured factors with selection[32,33]. Over-sampling of under-represented subgroups of the population is used, for example in the Millenium Cohort Study[34]. However, this solution will only remove bias due to selection into those specified subgroups (not any other selection bias). In addition, if the selection in those subgroups now differs according to other factors—e.g., the participators from the hard-to-reach groups are comparatively

healthier than those in the easier-to-recruit group, then new biases may be introduced.

A key advantage to the genetic analyses presented here over the observational analyses usually reported (and reported here in Table 1) is the ability to draw conclusions about causality (under the usual assumptions of MR, in particular, the assumptions around horizontal pleiotropy). For example, smoking is related to participation in the aide-memoire observationally (Table 1) but may be due to confounding as there is little evidence of an association using genetic variants associated with smoking (Supplementary Table S8). This information about causality may be useful to inform strategies to improve participation—for example, if smoking caused participation then qualitative work could be done to find out why smokers were less prone to participate, and then to address this in recruitment/retention strategies. However, if actually the association between smoking and participation is driven by (for example) socioeconomic position, and had nothing to do with their actual smoking, then targeting only smokers could be counter-productive. A strategy based only on improving participation in smokers could even induce more bias, in that interaction between socioeconomic position and smoking in their effect on participation might be induced.

There are a number of limitations to this analysis. First, our analysis sample was based on Europeans only in the UK Biobank sample. The UK Biobank is not population-representative and therefore these findings may not be generalisable to other population studies. Second, email access was only available at baseline and therefore this might not accurately reflect access to email at the release of the various optional components. Third, it is possible that some participants died before being able to participate in some of the optional components, however, this number will likely be small. Fourth, factors relating to participation may change with age. However, we saw strong genetic correlations with our UK Biobank participation measures and the ALSPAC measures. Fifth, the predictors used in MR, were selected a priori and it is possible we have missed some key predictors of participation. Finally, for MR we assume that the genetic variants used as an instrumental variable affect the outcome only through their effect on the exposure (i.e., the absence of horizontal pleiotropy). Our sensitivity analyses using MR-Egger and Median MR, which are more robust to horizontal pleiotropy were generally consistent, although often had much wider confidence intervals that crossed the null.

In summary, we demonstrate that genetic variants are associated with participation in several aspects of the UK Biobank study and that a wide range of traits cause differences in participation. This builds on previous work in the ALSPAC study and here we demonstrate strong genetic correlations between the UK Biobank participation measures and ALSPAC highlighting that these issues are likely to be seen in many studies. Our findings highlight the potential for introducing bias into both genetic and non-genetic analyses. All studies need to consider the importance of selection bias and use sensitivity analyses to assess the robustness of their conclusions.

## Methods

**UK Biobank**. This study was conducted using the UK Biobank resource, which has ethical approval and its own ethics committee (https://www.ukbiobank.ac.uk/ethics/). Details of the patient and public involvement in the UK Biobank are available online (www.ukbiobank.ac.uk/about-biobank-uk/ and https://www.ukbiobank.ac.uk/wp-content/uploads/2011/07/Summary-EGF-consultation.pdf?phpMyAdmin=trmKQlYdjjnQIgJ%2CfAzikMhEnx6). No patients were specifically involved in setting the research question or the outcome measures, nor were they involved in developing plans for recruitment, design, or implementation of this study. No patients were asked to advise on interpretation or writing up of results. There are no specific plans to disseminate the results of the research to study participants, but the UK Biobank disseminates key findings from projects on its website.

The UK Biobank study recruited over 500,000 individuals aged between 37 and 73 years (with >99.5% aged between 40 and 70 years) from across the UK between 2006 and 2010. The UK Biobank[35,36] collected extensive phenotypic and genotypic data on all participants. Here, we used data in up to 451,036 UK Biobank individuals who were defined as European ancestry using principal component analyses. Briefly, we generated principal components in the 100 Genomes Cohort, using high-confidence SMPs to obtain their individual loadings. The loadings were then used to project all of the UK Biobank samples into the same principal component space. The individuals were then clustered using principal components 1 to 4.

**Participation measures**. Four participation phenotypes were derived in the UK Biobank:

1. Percentage of food frequency questionnaires (FFQ) completed, based on the number of invites (data field 110002, 0–4) and the number of acceptances (data field 110001, 0–4). A binary variable was also created that represented sent a food frequency questionnaire but never accepted (0) and sent a food frequency questionnaire and completed at least one (1). This variable is based on the online requests which were sent out every 3–4 months a total of four times between February 2011 and June 2012 to participants who provided an email address at recruitment[37].

2. Participation in physical activity monitoring, a binary variable defined using data fields 110005 and 110006. 0 represents invited but not accepted and 1 represents invited and accepted. Between February 2013 and December 2015, a random sample of participants with a valid email was invited to wear the accelerometer. Participants from the North West region were excluded due to participant burden concerns[38].

3. Participation in a mental health questionnaire (MHQ), a binary variable defined using data fields 20400 and 20005. 0 represents invited but not accepted and 1 represents invited and accepted. Participants with a valid email were invited to complete the MHQ. The UK Biobank's contact approach was to (a) send an initial invitation email, (b) send a reminder email to non-responders (2 weeks after the initial invite), (c) send a reminder to partial responders (2 weeks after they started the questionnaire) and d) the last chance invitation after 4 months.

4. Aide-memoire completed, a binary variable derived from data field 111 which represents compliance to a request from the UK Biobank prior to attending the assessment centre to fill out specific information to help with the questionnaire. 0 represents non-compliance and 1 represents compliance.

With the exception of the aide-memoire, which was requested by everyone, the remaining variables relied on UK Biobank participants being invited to take part. The general UK Biobank protocol was to invite everyone to participate in the optional questionnaires and surveys, although as detailed above these invitations were generally sent via email. To investigate the impact of this strategy, we also created a variable to represent whether participants were invited to participate in at least one of the optional surveys above (coded as 1) or not (coded as 0).

**Genotypes**. We used imputed genotypes available from the UK Biobank for association analyses[39]. Variants were excluded if imputation quality (INFO) was <0.3 or the minor allele frequency (MAF) was <0.1%. This quality control process resulted in 6,930,712 variants for association analyses. Lead SNPs were defined as those with the smallest $P$ value and locus boundaries were defined using a ±0.5 Mb distance from the lead SNP.

**Observational associations**. Logistic regression analyses were used to explore the relationship between participant demographics and the four participation measures, plus the invitation measure. The Pearson correlations and overlap between the four participation measures were also investigated. Chi-squared analyses were used to explore the overlap of the binary participation measures.

**Genome-wide association analysis**. All individual variant association testing was performed using BOLT-LMM[40] v2.3. This software applies a linear mixed model (LMM) to adjust for population structure and individual relatedness. From the ~805,000 directly-genotyped (non-imputed) variants available, we identified 524,307 good-quality variants (bi-allelic SNPs; MAF ≥ 1%; HWE $P > 1 \times 10^{-6}$; non-missing in all genotype batches, total missingness <1.5% and not in a region of long-range LD) which BOLT-LMM used to build its relatedness model. A number of covariates (age, sex, UK Biobank assessment centre and genotyping platform (categorical; UKBiLEVE array, UKB Axiom array interim release and UKB Axiom array full release) were included at runtime. Here in the main paper, we only report variants that reached a stringent $P < 6 \times 10^{-9}$ cut-off based on simulations[41]. The results from the GWAS of receiving an invite to at least one of the three optional surveys is also reported.

**Genetic correlations**. We used a method based on LD score regression[42] as implemented in the LD Hub software[43], available at http://ldsc.broadinstitute.org/

ldhub/, to quantify the genetic overlap between the four participation traits and 832 traits with publicly available GWA data. This method uses the cross-products of summary test statistics from two GWASs and regresses them against a measure of how much variation each SNP tags (its LD score). Variants with high LD scores are more likely to contain more true signals and thus provide a greater chance of overlap with genuine signals between GWASs. Correlations were reported if they reached a Bonferroni corrected $P$ value (number of tests = 3220; $P < 1.5 \times 10^{-5}$).

We also used the LD score regression to explore the genetic correlation between our participation measures and those available in the ALSPAC study[1]. The LD score regression method used summary statistics from the GWAS meta-analysis of the 4 participation measures in UK Biobank and the participation measures of ALSPAC mother and children, calculates the cross-product of test statistics at each SNP, and then regresses the cross-product on the LD score.

Finally, we utilised LD score regression to explore the genetic correlation between not receiving an invite to participate in the various optional components and the four participation measures.

Genome-wide genetic correlations do not provide evidence of causality, which we tested with Mendelian randomisation using specific sets of variants. Instead, they likely represent a complex mixture of direct and indirect causal associations in both directions, pleiotropy and residual stratification. These likely properties of genome-wide genetic correlations mean they provide a way of projecting a phenotype measured in one study into another to test between study consistency (e.g., the ALSPAC versus UK Biobank comparison), or, when comparing different traits within one study, potentially a measure of correlation that is more representative of biological processes than observational correlations, although we note that observational correlations were usually very similar to genetic correlations.

**Mendelian randomisation**. We undertook two-sample MR analyses to further test the causal relationships between 80 exposure traits (decided a priori on the grounds that they are common exposures and used in current MR pipelines) (Supplementary Table 8) and the four different participation outcomes. The predictors were classified into nine broad categories (Supplementary Table 4).

The two-sample MR analyses used summary-level data from the BOLT-LMM GWAS of the participation traits. Known SNPs for each exposure trait (Supplementary Table 4) were extracted from the GWAS results to estimate the association of outcome and exposure-trait-SNP, whilst published coefficients from the primary GWAS were utilised for the association of exposure with exposure-trait-SNP. Four two-sample MR methods were performed using a custom pipeline: inverse-variance weighting (IVW); MR-Egger[24]; Weighted median (WM)[25]; Penalised weighted median (PWM)[25]. We have presented the IVW approach as our main analysis method, with the MR-Egger, WM and PWM representing sensitivity analyses to account for unidentified pleiotropy, which may bias our results. Horizontal pleiotropy occurs when the genetic variants related to the exposure of interest independently influence the outcome. IVW assumes there is either no horizontal pleiotropy under a fixed-effect model or, if using a random-effects model after detecting heterogeneity amongst the causal estimates, that the strength of the association between the genetic instruments and the exposure is not correlated with the magnitude of the pleiotropic effects (the InSIDE assumption) and that the pleiotropic effects have an average value of zero. MR-Egger estimates and adjusts for non-zero mean pleiotropy and therefore provides unbiased estimates if just the InSIDE assumption holds[24].

To explore the role of smoking heaviness on participation in the different smoking strata we performed one-sample MR in the unrelated subset of Europeans in the UK Biobank. We performed analyses in all individuals and stratified by smoking status into never, former, current and ever smokers. Here, for our binary participation measures, we first assessed the association between the cigarettes per day and the smoking GRS. The predicted values and residuals from this regression model were saved. Second, the predicted values from stage 1 were used as the independent variable and the participation measures as the dependent variable in a logistic regression model. As the FFQ participation measure was continuous we utilised the ivreg2 command in Stata.

All analyses were performed in Stata version 14 or R version 3.5.0.

**Reporting summary**. Further information on research design is available in the Nature Research Reporting Summary linked to this article.

## Data availability

This research has been conducted using the UK Biobank resource under application number 9072. The GWAS summary statistics generated in this study have been deposited in the GWAS catalogue (https://www.ebi.ac.uk/gwas/) under accession codes GCST90012790, GCST90012791, GCST90012792, GCST90012793, GCST90012794. All other data are available within the article or from the authors upon request.

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

## Acknowledgements

The authors would like to acknowledge the use of the University of Exeter High-Performance Computing (HPC) facility in carrying out this work. K.H. was funded by the Diabetes Research and Wellness Foundation (as part of a fellowship to J.T.). J.T. is supported by an Academy of Medical Sciences (AMS) Springboard award, which is supported by the AMS, the Wellcome Trust, GCRF, the Government Department of Business, Energy and Industrial strategy, the British Heart Foundation and Diabetes UK [SBF004\1079]. J.Z. is funded by a Vice-Chancellor Fellowship from the University of Bristol. This research was also funded by the UK Medical Research Council Integrative Epidemiology Unit (MC_UU_00011/1 and MC_UU_00011/4). This work was supported by the Integrative Epidemiology Unit, which receives funding from the UK Medical Research Council and the University of Bristol (MC_UU_00011/1, MC_UU_00011/3 and MC_UU_00011/4).

## Author contributions

J.T., G.D.-S., T.M.F. and K.T. contributed to the design of the study; J.T., J.Z., R.B., K.H., T.G.R., A.R.W., G.D.-S., T.M.F. and K.T. acquired, analysed and/or interpreted the data; J.T. G.D.-S., T.M.F. and K.T. drafted and/or made important contributions to the article and G.D.-S. and T.M.F. either provided technical or supervisory support for this work.

## Competing interests

The authors declare no competing interests.
