## [Peer Review File · Nature Communications]

Reviewers' comments:

Reviewer #1 (Remarks to the Author):

This is an important paper that addresses the increasing recognition of problems related to selection and ascertainment bias in the UK Biobank and similar datasets. The paper and approach are clear and analyses highly competent. I do, however, have some additional questions and suggestions for clarification.

Provide a better explanation of the four participation measures in the body of the paper. Although the measures are adequately described in the Methods section, the introduction of the measures and differences between them is limited and rather cryptic as it stands. This makes the results hard to understand and interpret at times, particularly when key measures like FFQ are introduced without an explanation of the acronym, etc. Or for instance on p. 4, you note that aide memoire captures a distinct aspect of participation but didn't outline clearly enough what it was in the first instance.

Strengthen the problem of selection, motivation of study and implications. On page 3, line 56-57 the authors rightly note problems with bias. Here it might also be useful to emphasise that selection has been shown to also influence the magnitude of association. Here I am thinking about the Keynes & Westreich (2019) article in the Lancet and related work that uses simulations to clarify this point ([https://www.thelancet.com/journals/lancet/article/PIIS0140-6736\(18\)33067-8/fulltext](https://www.thelancet.com/journals/lancet/article/PIIS0140-6736(18)33067-8/fulltext)). They show that whether an association is observed in one study is dependent upon the distribution of the exposure-outcome relationship in the discovery (e.g., UKBB) and target population. Here the point is slightly different in that they show that the magnitude of the association is thus highly dependent on the prevalence of other factors that interact with the exposure and outcome.

What is the overlap between the loci found for the 4 traits? Perhaps I missed it but I couldn't directly find whether this was reported. Even a simple Venn diagram in the Supplementary Material could be interesting to clarify the overlapping or unique loci. I assume you didn't run sex-specific GWAS since you don't have the power.

Interpretation of association of variants to additional traits sometimes lack reflection or interpretation. At times variants were reported as being linked to diseases or other traits with little explanation. For instance, on p. 7 line 124-5, 2 variants were linked to blood cell traits or 4 with more psychiatric or substance use traits. Or on p. 8, lines 156-163 with MHQ and APOE, where some link could have been made with early signals or lowered mental capacity to participate as well. Or, on p. 8, lines 166-68, we have an enigmatic association between aide memoire and bone mineral density. I agree that any biological interpretation would remain speculative since the authors don't (and arguably shouldn't) carry out any downstream biological annotation looking at enrichment or genes expressed at the protein level, etc. Given the distal nature of the phenotypes I doubt that work would be useful, but some discussion linking to other research could be helpful. Much of this is already in the detailed tables in the Supplementary Material.

Use of ALSPAC as comparative data. More reflection or motivation of why you use ALSPAC would be useful given the different study designs and populations. Beyond obvious ease of access to the data for the authors, substantive reasoning would be helpful such as the availability of comparative participation measures. I could not find reflection on the differences in study design, recruitment and population between UKBB and ALSPAC and how this might impact the results and comparison.

MR analyses. This is a strong aspect of the paper but at times that section lacked cohesion and a clear narrative of the main points. For example p. 11, lines 233-245 lists a variety of results and

the reader has to independently work to try to out the central points. The confidence intervals are really very wide for many of the estimates, so we need to be cautious but the authors do acknowledge this.

Emphasize relevance of the problem. The gravity of the problem could be clarified and more explicit in the paper and final discussion. For instance, on p. 12, those with higher breast cancer liability or psychological and neurological conditions had lower odds of participating. What are the consequences for interpretation or broader clinical concerns? In the discussion on page 15 the subtle but important point is made that conditioning on being invited to participate results in variables being biased towards the null rather than spurious associations, which is important. Perhaps a line using a disease or substantive example would bring this point home more forcefully. Given that the majority of these genetic studies have similar biases, collider bias will have the same impact so it raises concerns of how we can parse this out.

Strengthen solutions in discussion. Although the authors point to some brief solutions on p. 16, line 347 of inverse probability weighting or multiple imputation, this could be strengthened. Also, some reference to work that has already developed these kinds of weights could be useful (e.g., for mortality selection in HRS <https://www.ncbi.nlm.nih.gov/pubmed/28402496>). Here I also miss the obvious suggestion to recruit and oversample populations from lower socioeconomic groups, less healthy, non-European ancestry, since we know this is increasingly important. Also, the recognition of variable prediction within ancestry groups related to socioeconomic status and so forth (<https://elifesciences.org/articles/48376>).

Finally, this likely goes beyond the auspices of this paper but I do suspect that there is a broader underlying latent factor or cause that influences participation *and* some of the core observed genetic correlations such as higher educational attainment, intelligence, risk and so forth. It is likely driven by socioeconomic status or a general altruistic latent factor (or the luxury to be able to be altruistic). This likely takes the paper too far in another direction, but this idea could be explored using Genomic SEM (structural equation modelling) (<https://www.ncbi.nlm.nih.gov/pubmed/30962613>). Here you could fit some additional models on both participation and for example years of education to test for mediating traits to see if the genetic correlations are independent from these factors. It wouldn't get at causality the way the MR models do, but it might help thinking about whether the genetic correlations are directly related to the coupling of these traits (participation, education and intelligence) or downstream of some sort of common identified latent cause.

Minor Points

Some references to the MR methods would be useful in the body of the paper.

Figures. Some of the abbreviated terms need to be described in notes

Figure 1. Although I like the Venn diagrams, not sure how informative it is using this metric. It takes some time to digest it.

Figure 2. I wonder if the figure is useful or informative enough to be included in the main body of the paper.

Figure 3, plot A – could you combine education and intelligence into one graph? Seems like considerable overlap

Melinda C. Mills

Reviewer #2 (Remarks to the Author):

In this paper, the authors explore the effect of several phenotypes on participation in genetic studies. I think the paper is certainly of great interest for the scientific community, in particular for

researchers involved in large genetic studies, as it gives hints for study design and gives warnings for the interpretation of GWAS results.

However, in my opinion the manuscript is hard to read and I think that it needs improvements in particular in terms of presentation of the results.

Major comments:

1. In the introduction the authors mention that UKBiobank has several measures of participation. It's not clear why they focused on these four optional components in study.

2. The first part of the Results section is a bit difficult to read. I suggest the authors to clarify at the beginning which components they study and to define the abbreviations they will use in the rest of the text.

3. Participation was associated with many traits listed in the Results. Is that list comprehensive? How many (and which) traits did they test in total?

4. Do the 4 variants associated with FFQ and in LD with ADHD-associated variants include the 2 variants associated with intelligence and cognitive performance?

5. It's really good to see that the loci associated with MHQ identified by Adam et al replicated here. Which is the pvalue in Adam et al for the 6 additional variants found in the current study? Are the summary stats publicly available to check that?

6. The authors calculated the genetic correlation between the participation and GWAS and ALSPAC. Where are those results? They should list them in a Supplementary Table

7. In the Mendelian Randomisation section the authors say "Higher BMI caused lower odds of participation in the FFQ and physical activity monitoring in women only when the 72 BMI variants were considered"

This sentence is not clear to me. Why do the authors specify the number of variants used in the MR model? why do they specify it only for BMI? Did they conduct MR analysis for BMI using also a different of SNPs?

We would like to thank the Reviewer's for their helpful comments and believe the revised manuscript is significantly improved. We have provided a detailed point by point response to Review below:

Reviewers' comments:

Reviewer #1 (Remarks to the Author):

This is an important paper that addresses the increasing recognition of problems related to selection and ascertainment bias in the UK Biobank and similar datasets. The paper and approach are clear and analyses highly competent. I do, however, have some additional questions and suggestions for clarification.

Provide a better explanation of the four participation measures in the body of the paper.

Although the measures are adequately described in the Methods section, the introduction of the measures and differences between them is limited and rather cryptic as it stands. This makes the results hard to understand and interpret at times, particularly when key measures like FFQ are introduced without an explanation of the acronym, etc. Or for instance on p. 4, you note that aide memoire captures a distinct aspect of participation but didn't outline clearly enough what it was in the first instance.

Response: We have amended the methods to provide more detail about the four participation measures and included some additional text in the results to remind the reader about the different optional components of the UK Biobank. The final paragraph of the introduction now includes the following statement: "The four optional components tested were a) the percentage of food frequency questionnaires (FFQ) completed, b) acceptance of the invite to wear a physical activity monitor, c) acceptance of an invite to participate in the mental health questionnaire (MHQ) and d) the completion of the aide memoire.". Additional edits have been made in the results and methods, including ensuring that any abbreviations used are defined. We have also added to the

discussion some more information about the aide memoire: “The fourth participation measure considered was the aide memoire, where participants were asked at baseline to complete a short form with specific data. Our analyses suggest that this measure captures another aspect of behaviour, perhaps reflecting compliance rather than participation, with evidence that a genetic liability to riskier behaviour was inversely associated with completing the aide memoire.”

Strengthen the problem of selection, motivation of study and implications. On page 3, line 56-57 the authors rightly note problems with bias. Here it might also be useful to emphasise that selection has been shown to also influence the magnitude of association. Here I am thinking about the Keynes & Westreich (2019) article in the Lancet and related work that uses simulations to clarify this point

([https://www.thelancet.com/journals/lancet/article/PIIS0140-6736\(18\)33067-8/fulltext](https://www.thelancet.com/journals/lancet/article/PIIS0140-6736(18)33067-8/fulltext)). They show that whether an association is observed in one study is dependent upon the distribution of the exposure-outcome relationship in the discovery (e.g., UKBB) and target population. Here the point is slightly different in that they show that the magnitude of the association is thus highly dependent on the prevalence of other factors that interact with the exposure and outcome.

Response: We agree that there are many circumstances where selection can lead to bias. We have added the following sentence to the introduction “ . As another example, selection bias can occur if a modifier of the effect of exposure on outcome causes selection. A comparison of associations between risk factors and overall and cause-specific mortality in UK Biobank and the less-selected Health Survey for England and Scottish Health Surveys showed wide variation in these associations, with some over-estimated in UK Biobank and some under-estimated.”

What is the overlap between the loci found for the 4 traits? Perhaps I missed it but I couldn't directly find whether this was reported. Even a simple Venn diagram in the Supplementary Material could be interesting to clarify the overlapping or unique loci. I assume you didn't run sex-specific GWAS since you don't have the power.

Response: Table 2 and ST5 provides details of other participation measures with variants within 500kb of the identified genome wide variant. We have now updated this to include the R^2 values for the variants within 500kb and added some text to the results as follows: “Many of the lead variants were within 500kb of another lead variant for a different

participation measure (Table 2 and Supplementary table 5). For example, 6/8 of the FFQ lead variants at $P < 6 \times 10^{-9}$ were within 500kb of another lead variant for either actigraphy or MHQ, whilst the only variant at $P < 6 \times 10^{-9}$ for actigraphy was within 500kb of a FFQ and MHQ variant and 4/21 variants at $P < 6 \times 10^{-9}$ for the MHQ were within 500kb of a FFQ or actigraphy variant.”

Interpretation of association of variants to additional traits sometimes lack reflection or interpretation. At times variants were reported as being linked to diseases or other traits with little explanation. For instance, on p. 7 line 124-5, 2 variants were linked to blood cell traits or 4 with more psychiatric or substance use traits. Or on p. 8, lines 156-163 with MHQ and APOE, where some link could have been made with early signals or lowered mental capacity to participate as well. Or, on p. 8, lines 166-68, we have an enigmatic association between aide memoire and bone mineral density. I agree that any biological interpretation would remain speculative since the authors don't (and arguably shouldn't) carry out any downstream biological annotation looking at enrichment or genes expressed at the protein level, etc. Given the distal nature of the phenotypes I doubt that work would be useful, but some discussion linking to other research could be helpful. Much of this is already in the detailed tables in the Supplementary Material.

Response: We agree with the reviewer that some more reflection around the association of variants with additional traits would be helpful and have updated the manuscript to reflect this – with more information in the results section. For example, following the APOE results we have now included: “This suggests that individuals with early signs of cognitive impairment lowered an individual’s capacity to participate in the MHQ.” Similarly following the link between the aide memoire variant and bone mineral density we have now added: “The G allele raises both isoleucine levels and bone mineral density (BMD) and was associated with lower odds of completing the aide memoire. Previous studies have demonstrated that BMD is inversely associated with cognition and Alzheimer’s disease, indicating those with higher BMD may have a better memory.”

Use of ALSPAC as comparative data. More reflection or motivation of why you use ALSPAC would be useful given the different study designs and populations. Beyond obvious ease of access to the data for the authors, substantive reasoning would be helpful such as the availability of comparative participation measures. I could not find reflection on the

differences in study design, recruitment and population between UKBB and ALSPAC and how this might impact the results and comparison.

Response: The genetic correlations to compare participation measures in ALSPAC and UKB were carried out to demonstrate that the problem is not specific to one study. The strong genetic correlations between the participation measures in 2 distinct and methodological different studies highlight this. We have added some text to the end of the introduction to summarise why these genetic correlations were important: “ If the same factors affect participation in studies that vary by geography, time period and design, then those studies will suffer the same bias, and thus replication of results across studies becomes meaningless.” Further in the discussion we have amended the paragraph about these results to read: “These results suggest that similar genetic factors are driving participation in follow-up and optional components of studies, regardless of study design, recruitment strategies and the population demographics of the study. Similarity of factors affecting participation across different studies is potentially important for comparisons of results between studies – if similar factors cause participation in different studies, then collider bias will have the same impact on the results from each study. Thus, results from different studies would be subject to similar biases, causing replication of results across studies to become meaningless.”

MR analyses. This is a strong aspect of the paper but at times that section lacked cohesion and a clear narrative of the main points. For example p. 11, lines 233-245 lists a variety of results and the reader has to independently work to try to out the central points. The confidence intervals are really very wide for many of the estimates, so we need to be cautious but the authors do acknowledge this.

Response: We have added some short headers to more clearly highlight the main MR messages presented on pages 11-13.

Emphasize relevance of the problem. The gravity of the problem could be clarified and more explicit in the paper and final discussion. For instance, on p. 12, those with higher breast cancer liability or psychological and neurological conditions had lower odds of participating. What are the consequences for interpretation or broader clinical concerns? In the discussion on page 15 the subtle but important point is made that conditioning on being invited to participate results in variables being biased towards the null rather than spurious associations, which is important. Perhaps a line using a disease or substantive example

would bring this point home more forcefully. Given that the majority of these genetic studies have similar biases, collider bias will have the same impact so it raises concerns of how we can parse this out.

Response: Our point about conditioning on being invited to participate not resulting in spurious associations was particularly referring to our analysis of factors associated with participation (which are conditional on being invited to participate), and was reflecting only on those variables which are in truth positively associated with participation. We have clarified this:

“This suggests that here conditioning on being invited to participate could have resulted in the Mendelian randomisation analyses for these variables being biased towards the null, rather than inducing spurious associations if they were in truth positively associated with participation. However, if there are non-linearities or interactions in the effects of the risk factors on invitation/participation, then the direction of the bias cannot be predicted. Similarly, a factor that affects being invited, but does not in truth affect participation, could appear to have a positive or negative spurious association with participation, conditional on being invited.”

We have also added broader discussion of impacts to the discussion:

“Selection of the type demonstrated here may cause bias in estimates of effects, and the size and direction of bias cannot (usually) be exactly determined. Previous work has shown that estimated effects of risk factors on mortality and cause-specific mortality differ between UK Biobank and the less-selected Health Survey for England – with some being moved towards, and some away from the null. This could imply that selection into UK Biobank is causing bias in estimating these effects. For example, we have shown that smoking is negatively associated with participation in UK Biobank. If a factor that causes lung cancer is also negatively associated with participation (e.g. socioeconomic position), then selecting on participating in UK Biobank would induce a negative association between smoking and lung cancer (assuming an additive model), which would bias the estimated effect of smoking on lung cancer towards the null. This is indeed what is seen in the comparison of estimates for this effect between UK Biobank and HSE-SHS. It should be noted that this simple estimate of the direction of bias depends on assumptions about the underlying selection model, and cannot be verified with only UK Biobank data – e.g.

an interaction between smoking and socioeconomic position in their effect on participation could change the size and direction of any bias. We have similarly shown previously that some effect estimates were different when calculated on only those continuing to participate in ALSPAC, compared to all those participating at baseline. It has also been suggested that selection bias may (at least in part) be responsible for overestimates of the protective effect of moderate alcohol consumption.”

Strengthen solutions in discussion. Although the authors point to some brief solutions on p. 16, line 347 of inverse probability weighting or multiple imputation, this could be strengthened. Also, some reference to work that has already developed these kinds of weights could be useful (e.g., for mortality selection in

HRS <https://www.ncbi.nlm.nih.gov/pubmed/28402496>

). Here I also miss the obvious suggestion to recruit and oversample populations from lower socioeconomic groups, less healthy, non-European ancestry, since we know this is increasingly important. Also, the recognition of variable prediction within ancestry groups related to socioeconomic status and so forth (<https://elifesciences.org/articles/48376>).

Response: We have added a new paragraph to the discussion relating to potential solutions.

“Strategies to investigate or minimise the impact of selection on a given estimate depend on the data available on the population not selected into the study. Inverse probability weighting (IPW) has been suggested to overcome mortality bias, but the validity of this depends on correctly specifying the selection model. If there is an unmeasured factor that affects selection, and is related to the variables in the analysis model, then this may mean that inverse probability weighting is not unbiased. IPW as a solution also depends on having data on all the variables affecting selection and their distribution in the population in which we wish to make inference. Solutions using IPW to infer bounds on estimates have been proposed, although these can result in wide bounds, or depend on underlying assumptions about associations of unmeasured factors with selection. Over-sampling of under-represented subgroups of the population is used, for example in the Millenium Cohort Study. However, this solution will only remove bias due to selection into those specified subgroups (not any other selection bias). In addition, if selection in those subgroups now differs according to other factors – e.g. the participators from the

hard-to-reach groups are comparatively healthier than those in the easier-to-recruit group, then new biases may be introduced.”

Finally, this likely goes beyond the auspices of this paper but I do suspect that there is a broader underlying latent factor or cause that influences participation *and* some of the core observed genetic correlations such as higher educational attainment, intelligence, risk and so forth. It is likely driven by socioeconomic status or a general altruistic latent factor (or the luxury to be able to be altruistic). This likely takes the paper too far in another direction, but this idea could be explored using Genomic SEM (structural equation modelling)

(<https://www.ncbi.nlm.nih.gov/pubmed/30962613>

). Here you could fit some additional models on both participation and for example years of education to test for mediating traits to see if the genetic correlations are independent from these factors. It wouldn't get at causality the way the MR models do, but it might help thinking about whether the genetic correlations are directly related to the coupling of these traits (participation, education and intelligence) or downstream of some sort of common identified latent cause.

Response: We agree with the reviewer that there may be underlying latent factors like socioeconomic status or a general altruistic latent factor, but we believe this is beyond the scope of this paper and would warrant a paper in its own right.

Minor Points

Some references to the MR methods would be useful in the body of the paper.

Response: We have added some relevant MR methods references into the main body of the paper.

Figures. Some of the abbreviated terms need to be described in notes

Figure 1. Although I like the Venn diagrams, not sure how informative it is using this metric. It takes some time to digest it.

Response: We have removed the Venn diagrams from the manuscript.

Figure 2. I wonder if the figure is useful or informative enough to be included in the main body of the paper.

Response: Based on the other Reviewer's comments we have now included these correlation results as a supplementary table [Supplementary table 2].

Figure 3, plot A – could you combine education and intelligence into one graph? Seems like considerable overlap

Response: The issue with combining onto one graph is the different y-axis as the effects are much smaller for the intelligence analyses and when plotted on one graph it is impossible to see the intelligence relationship with participation.

Melinda C. Mills

Reviewer #2 (Remarks to the Author):

In this paper, the authors explore the effect of several phenotypes on participation in genetic studies. I think the paper is certainly of great interest for the scientific community, in particular for researchers involved in large genetic studies, as it gives hints for study design and gives warnings for the interpretation of GWAS results.

However, in my opinion the manuscript is hard to read and I think that it needs improvements in particular in terms of presentation of the results.

Major comments:

1. In the introduction the authors mention that UKBiobank has several measures of participation. It's not clear why they focused on these four optional components in study.

Response: We focussed on the four components for which data on participation was available. Data on other optional components in the UK Biobank including attending imaging and completing the online occupational questionnaire are not currently available to researchers. We have clarified this in the introduction, by stating: "The four optional components tested were a) the percentage of food frequency questionnaires (FFQ) completed, b) acceptance of the invite to wear a physical activity monitor, c) acceptance of an invite to participate in the mental health questionnaire (MHQ) and d) the completion of the aide memoire."

2. The first part of the Results section is a bit difficult to read. I suggest the authors to clarify at the beginning which components they study and to define the abbreviations they will use in the rest of the text.

Response: As mentioned in the response above the end of the introduction now introduces the four participation measures. We have also improved the clarity of the results section,

ensuring that all abbreviations are defined and used in the remainder of the text.

3. Participation was associated with many traits listed in the Results. Is that list comprehensive? How many (and which) traits did they test in total?

Response: In the Mendelian randomisation section of the methods we discuss the 80 predictor traits used (summarised in ST8) – these were decided on a priori based on the fact that they were a) common exposures, b) there was some previous evidence of potential involvement in participation from previous studies and genetic correlations and c) they were available in current MR pipelines. We acknowledge this is not an exhaustive list and have added a brief sentence to the limitations about this: “Fifth, the predictors used in MR, were selected a priori and it is possible we have missed some key predictors of participation.”.

4. Do the 4 variants associated with FFQ and in LD with ADHD-associated variants include the 2 variants associated with intelligence and cognitive performance?

Response: Yes, the variants in LD with ADHD variants are also known intelligence loci or in LD with these loci. The columns “Known association signal” and “GWAS hits for loci in high LD ($r^2 > 0.8$)” provide further information on those associations, but we have clarified this in the main manuscript as well.

5. It's really good to see that the loci associated with MHQ identified by Adam et al replicated here. Which is the pvalue in Adam et al for the 6 additional variants found in the current study? Are the summary stats publicly available to check that?

Response: We have accessed the summary statistics here: <https://datashare.is.ed.ac.uk/handle/10283/3335> and now included the results from this analysis in Table 2. Briefly, the 6 additional variants identified at the stringent P value cut off were all directionally consistent with $p\text{-values} < 1 \times 10^{-5}$.

6. The authors calculated the genetic correlation between the participation and GWAS and ALSPAC . Where are those results? They should list them in a Supplementary Table

Response: We had previously only included the correlation statistics between the ALSPAC measures and the UKB measures in the main results text and as a figure. We have now added these results to Supplementary table 2.

7. In the Mendelian Randomisation section the authors say "Higher BMI caused lower odds

of participation in the FFQ and physical activity monitoring in women only when the 72 BMI variants were considered"

This sentence is not clear to me. Why do the authors specify the number of variants used in the MR model? why do they specify it only for BMI? Did they conduct MR analysis for BMI using also a different of SNPs?

Response: We apologise for the lack of clarity here, in an earlier version of the manuscript we had included the 72 Locke et al BMI variants and the larger set of 942 variants more recently identified by the GIANT consortium. We subsequently removed these as they are subjected to winner's curse. We have updated the manuscript to alter the wording and remove the 72 SNPs component.

REVIEWERS' COMMENTS:

Reviewer #1 (Remarks to the Author):

The authors have now amended both the final paragraph of the introduction and the methods to clarify their key measures, which is a great improvement for the clarity of the article.

I am satisfied that they now explicitly clarify in the introduction that selection can lead to bias in many circumstances, which strengthens the motivation for their study. I appreciate the use of a concrete example as well.

I asked about the overlap of loci found for the 4 traits. Table 2 and Supp Tab 5 now include the R² values and within 500kb and explicitly acknowledge the overlap in the text. There is indeed quite some overlap for some with 6/8 of the FFQ lead variants within 500kb for a lead variant for actigraphy or MHQ.

It is good that they now add more reflection and interpretation of the association of variants to additional traits and it is no as cryptic.

The rewrite of the MR section on pages 11-13 is now better and highlights the main messages more explicitly.

The additions and revisions to the conclusion and final discussion are now strengthened and it is excellent that they have added a statement about the broader impact and relevance of the problem.

Beyond only identifying the problem I am also pleased to see they took up the challenge to try to discuss solutions such as inverse probability weighting and the inherent biases with that and other potential solutions.

The authors explicitly addressed all of my concerns. The message and clarity of the manuscript has now been greatly improved. I have no further suggestions for revisions.

Reviewer #2 (Remarks to the Author):

Thank you to the authors for addressing my points. I think that now the results are clear. I do like this paper.

REVIEWERS' COMMENTS:

Reviewer #1 (Remarks to the Author):

The authors have now amended both the final paragraph of the introduction and the methods to clarify their key measures, which is a great improvement for the clarity of the article.

I am satisfied that they now explicitly clarify in the introduction that selection can lead to bias in many circumstances, which strengthens the motivation for their study. I appreciate the use of a concrete example as well.

I asked about the overlap of loci found for the 4 traits. Table 2 and Supp Tab 5 now include the R² values and within 500kb and explicitly acknowledge the overlap in the text. There is indeed quite some overlap for some with 6/8 of the FFQ lead variants within 500kb for a lead variant for actigraphy or MHQ.

It is good that they now add more reflection and interpretation of the association of variants to additional traits and it is no as cryptic.

The rewrite of the MR section on pages 11-13 is now better and highlights the main messages more explicitly.

The additions and revisions to the conclusion and final discussion are now strengthened and it is excellent that they have added a statement about the broader impact and relevance of the problem.

Beyond only identifying the problem I am also pleased to see they took up the challenge to try to discuss solutions such as inverse probability weighting and the inherent biases with that and other potential solutions.

The authors explicitly addressed all of my concerns. The message and clarity of the manuscript has now been greatly improved. I have no further suggestions for revisions.

Reviewer #2 (Remarks to the Author):

Thank you to the authors for addressing my points. I think that now the results are clear. I do like this paper.